# Identifying where Japanese agriculture is most at risk: A longitudinal analytical framework based on municipal boundaries as of 1950 for workforce decline and aging (2005–2020)

**Kazuho Funakawa**[1]*, **Toshihiro Sakamoto**[1], **Kohei Imamura**[1], **Mizuki Morishita**[1], **Shoji Taniguchi**[2], **Nobusuke Iwasaki**[3], **Gen Sakurai**[1]

**1** Institute for Agro-Environmental Sciences, National Agricultural and Food Research Organization, Tsukuba, Ibaraki, Japan, **2** Research Center for Agricultural Information Technology, National Agricultural and Food Research Organization, Tsukuba, Ibaraki, Japan, **3** Faculty of Agriculture, Tottori University, Tottori, Tottori, Japan

* funakawa.kazuho041@naro.go.jp

## Abstract

Japan's agricultural workforce is shrinking and aging, posing a significant social issue. Yet, fundamental analysis—quantifying and mapping where and how rapidly this demographic shift is progressing—have been lacking, largely due to extensive municipal boundary reorganizations in Japan. This study aimed to visualize and clarify the current demographic shifts by restoring temporal comparability through a stable spatial baseline: the "sub-municipalities." Using Census of Agriculture and Forestry data for 2005, 2010, 2015, and 2020, we mapped two indicators for core agricultural workers: (i) the decline rate between 2005 and 2020, and (ii) the proportion of workers aged 75 years and above in 2020, revealing the regional landscape of workforce shrinkage and aging. We produced nationwide maps at the sub-municipality level, summarized land-type trends using generalized mixed models, and identified areas of extreme change using hotspot/coldspot analysis. The results revealed a nationwide downturn with pronounced spatial heterogeneity: the strongest declines were observed in mountainous and upland-dominated areas, whereas they tended to be more moderate in flatland paddy areas. Hotspots were scattered throughout the country, but were mainly located in areas with significant geographical constraints. Coldspots, representing modest growth, were identified around the Kinki and northern Kita-Kyushu metropolitan fringes. The aging rate was the highest in mountainous and paddy areas, whereas flatland and upland-dominated areas tended to be more resilient in this aspect. Aging hotspots aligned with the Tokai–Tosan Mountain belt and the Sanyo and San'in regions, whereas coldspots were observed in Hokkaido and Tohoku regions. Although we focused on the numerical and age composition of core agricultural workers, the approach can be generalized to other census indicators

**Data availability statement:** The processed datasets underlying the results presented in this study are available in the Zenodo repository at https://zenodo.org/records/17196659 (DOI: 10.5281/zenodo.17196659). The original raw data were obtained from the Statistics Bureau of Japan via the e-Stat portal (https://www.e-stat.go.jp/), including the Agricultural Census (Ministry of Agriculture, Forestry and Fisheries), and the National Land Numerical Information (Ministry of Land, Infrastructure, Transport and Tourism). These raw data are publicly available under the Government of Japan Standard Terms of Use.

**Funding:** This work was supported by the Environmental Restoration and Conservation Agency (grant number JPMEERF25S12421) awarded to G.S., N.I., T.S., M.M., K.I., and S.T.(https://www.erca.go.jp/erca/english/index.html). Additional support was provided by the Japan Society for the Promotion of Science (grant number 25K21880) awarded to N.I. (https://www.jsps.go.jp/english/). The funders had no role in study design, data collection and analysis, decision to publish, or preparation of the manuscript.

**Competing interests:** The authors have declared that no competing interests exist.

(e.g., sales, cultivated area, and production type), supporting locally adapted, evidence-based rural policy.

## Introduction

In recent years, the aging of the agricultural workforce has emerged as a serious social issue in many developed countries [1–5], accompanied by a general decline in the number of people engaged in farming [6,7]. As a consequence, the sustainability of agricultural production and rural communities faces growing concerns, which have drawn attention from policymakers and researchers [8–11].

In Japan, these trends are even more acute than in many other developed countries [12–14]. Between 2005 and 2020, the number of core agricultural workers (Farmers primarily engaged in agriculture, as defined by the Ministry of Agriculture, Forestry and Fisheries [MAFF]) decreased from approximately 2,241,000–1,363,000, a sharp decline of 39.2%. Meanwhile, the proportion of agricultural workers aged 65 years and above rose from 57.4% in 2005 to 69.6% in 2020, highlighting the rapid aging of the agricultural labor force [15,16].

The decline and aging of the agricultural workforce have triggered a cascade of secondary problems, such as the expansion of farmland abandonment [17,18], shrinkage and weakening of rural communities [19,20], and deterioration of land management in mountains, forests, and river basins [21,22]. Consequently, concerns regarding an increase in natural disasters such as landslides and flooding [23,24], as well as greater damage caused by wild animals encroaching into human-inhabited areas [25,26], are growing.

These problems are not uniformly distributed across the country, as the effects of agricultural population decline and aging are strongly influenced by area-specific geographical and socio-economic conditions [18,27,28]. Factors such as proximity to consumer markets (urban centers), the degree of farmland consolidation, topography, and climate can affect the severity and nature of the challenges faced by local agricultural communities. To design effective responses to these agricultural issues, it is essential to account for this heterogeneity and to quantitatively assess the extent and pace of population decline and aging across each region nationwide.

However, most previous studies focused on broader spatial scales, such as the national scale [29,30], agricultural policy blocks (10–14 areas) [31], or prefectures (47 areas) [32,33], whereas detailed analyses at the municipal scale across the country are relatively limited. A key reason why quantitative municipal-level analyses have been scarce lies in the large-scale reorganization of municipalities between 1999 and 2010, known as the Great Merger of Municipalities during the Heisei era. During this period, the number of municipalities in Japan declined dramatically from 3,229–1,821 [34]. This restructuring involved widespread integration, division, and modification of municipal boundaries, which complicated longitudinal analyses based on municipalities. Therefore, studies using agricultural statistics at the scale of municipalities or smaller spatial units face considerable technical obstacles, particularly those aiming to provide a comprehensive overview of the Japanese archipelago.

To fill this methodological gap, we established a framework for longitudinal analysis by standardizing administrative codes over time based on the years in which the Japanese Census of Agriculture and Forestry was conducted. Specifically, we utilized the municipal boundaries defined on February 1st, 1950, which represent units significantly smaller than present-day municipalities. Hereafter, these units are referred to as "sub-municipalities" in this study. By adopting the sub-municipalities as the fundamental spatial units, this framework minimizes the effects of subsequent boundary mergers, divisions, and modifications on the definition of spatial units, allowing more reliable tracking of temporal changes. This framework thus enables the consistent integration of agricultural statistical data across periods and spatial scales.

The objective of this study was to visualize and clarify temporal demographic shifts in Japanese agriculture, focusing on population decline and aging, and their spatial characteristics at a fine-grained local scale. By quantitatively mapping these trends across sub-municipalities, we provided the first nationwide identification of spatial clusters where decline and aging are most pronounced, thereby offering essential baseline information for agricultural research and contributing to more nuanced, locally adapted policy discussions.

## Data and methods

### Data source: The census of agriculture and forestry in Japan

This study used data from the Census of Agriculture and Forestry, a nationwide survey conducted every five years by the MAFF. The census is a fundamental statistical survey mandated by the Statistics Act, which aims to capture comprehensive information on the structure and dynamics of agriculture in Japan. For the analysis, census data from four survey years, 2005, 2010, 2015, and 2020, were obtained from the official e-Stat portal managed by the Statistics Bureau of Japan [35]. The e-Stat is the official portal site for government statistics in Japan, providing access to view and download statistical data compiled by the Japanese government. Since a major revision in data collection was implemented in 2005, which altered the statistical definitions and coverage, continuous and comparable datasets have been available from that year onward.

Among the various indicators included in the census, this study focused on the number and demographic composition of core agricultural workers (kikan-teki nōgyō jūjisha). According to the MAFF's definition, core agricultural workers are individuals whose usual status is that of a working adult (i.e., not a student or homemaker), and whose principal occupation is farming rather than non-agricultural activities such as retail or wage employment. They represent the major labor force directly responsible for agricultural production.

This category includes only individuals from individual farm management entities (i.e., family-run farms) and excludes those employed by organizational management entities, such as agricultural corporations or cooperatives. In the 2005–2015 censuses, incorporated family farms were still classified as individual entities, whereas in the 2020 census, these were reclassified as organizational entities. While this change introduces a small inconsistency in the scope of the target population, its impact is minimal. For example, incorporated family farms constituted only 0.3% of all individual entities in 2015, and 0.6% in 2020. Therefore, to calculate trends in workforce size and aging rates, this study treated the number of core agricultural workers under individual entities in 2020 as comparable to those in previous years.

### Hierarchical spatial units and administrative codes in the census of agriculture and forestry

The Census of Agriculture and Forestry of Japan employs a four-tiered hierarchy of spatial units to organize statistical data: (1) prefectures, (2) municipalities, (3) municipalities as of 1950 (sub-municipalities), and (4) agricultural settlements. Each unit was assigned a two- to three-digit administrative code, and an overall identifier for any spatial unit was constructed as a 10-digit code by concatenating the codes from all four tiers. The codes were nested, meaning that each subunit's code was sequentially numbered within its parent unit. For example, the code "0822006010" represents Ibaraki (08, prefecture), Tsukuba City (220, municipality), Onogawa village (06, sub-municipality), and Nojo (010, agricultural

settlement). As a result of this hierarchical structure, when a municipal merger occurs, all eight digits representing the sub-prefecture levels are reassigned.

The prefectural boundaries have remained virtually unchanged since 2005, except for a few minor modifications. Similarly, the boundaries of the approximately 11,000 sub-municipalities have remained remarkably stable, with only a few dozen changes since 2005. In contrast, municipalities were substantially reorganized during the Great Merger of Municipalities in the Heisei era, resulting in boundary changes for nearly half of them. Agricultural settlements, the smallest unit, are unsuitable for frequent redefinition with each census iteration. The lack of spatial continuity at the municipal and settlement levels prevents consistent comparison across survey years.

Given these limitations, this study adopted the sub-municipalities as the spatial unit for longitudinal analysis, as they provided a suitable balance between spatial resolution and boundary stability, enabling consistent tracking of demographic and agricultural trends over time. Although the geographical boundaries of sub-municipalities have remained largely unchanged since 1950, their administrative codes have undergone repeated revisions, resulting in inconsistencies across survey years. Consequently, agricultural statistics at the sub-municipality scale have not been quantitatively analyzed or mapped on a nationwide basis. To address this issue, the codes were standardized through the following procedure.

### Tracking of temporal changes in administrative codes

To ensure consistency across survey years, we constructed a correspondence table that traces changes in administrative codes over time. This involved systematically identifying how each unit's 10-digit code evolved because of boundary adjustments or name changes.

Merger histories were obtained from official records published by the Ministry of Internal Affairs and Communications [36]. Based on these records, we determined the official municipal names for each census reference year, 2005, 2010, 2015, and 2020, and established a one-to-one correspondence for merged municipalities. These correspondences were then linked to the sub-municipalities defined in the census data to reconstruct the temporal transitions of administrative codes. In cases where a sub-municipality was divided among multiple reestablished municipalities, its code was marked as missing (NA) to prevent erroneous attribution. Minor non-structural changes, such as character replacements or the reordering of place names, were manually corrected to maintain continuity. The resulting dataset provided a consistent basis for longitudinal analysis at the sub-municipality level.

### Analytical indicators and statistical approaches

This study employed two key indicators to assess the current state of Japanese agriculture: the decline rate and the proportion of the elderly among total core agricultural workers. The decline rate was defined as the proportional decrease in the number of core agricultural workers between 2005 and 2020. The proportion of the elderly was defined as the proportion of workers aged 75 years or above among all core agricultural workers in a given census year. While 65 years is commonly used as a threshold for defining the elderly, this study adopted a 75-year threshold, considering Japan's uniquely rapid demographic aging. Given that a substantial portion of the agricultural workforce is already over 65 years of age, this stricter criterion was deemed more appropriate for capturing the degree of aging in the Japanese agricultural sector. For reference, using the same composition-of-workers metric at the economy-wide level, workers aged ≥75 accounted for 3.52% of all employed persons in Japan, based on the 2020 Population Census [37].

To examine how these indicators vary across agricultural land types, we constructed generalized linear mixed models (GLMMs) with a beta distribution with a logit link function, as the dependent variables were bounded between 0 and 1. In the model concerning the decline rate of the agricultural labor force, the dependent variable was defined as the ratio of core agricultural workers in 2020 to those in 2005, whereas in the model concerning the aging rate, it was defined as the proportion of elderly workers in 2020. Accordingly, the former analysis captures regional differences in changes that occurred between 2005 and 2020, while the latter explains regional differences observed in 2020. The GLMM framework

was chosen to account for both the proportional nature of the response variables and the hierarchical structure of the data. Specifically, fixed effects included the combination of topographical categories (Urban, Flatland, Hilly, and Mountainous Agricultural Area; landtype1) and land-use types (Paddy-Dominated, Paddy-Upland-Mixed, and Upland-Dominated Type; landtype2), as defined by the MAFF [38], forming a nested classification structure (Fig 1). Full definitions, indicator descriptions, and threshold rules for the MAFF agricultural area classification (landtype1 and landtype2) are provided in S1 Appendix (S1 File). Additionally, 14 regional blocks (Fig 1) were included as random effects to account for spatial heterogeneity and potential clustering effects that may influence the demographic trends independently of land type. For the decline rate model, sub-municipalities that showed a net increase in the number of core agricultural workers (i.e., those with negative decline rates) were excluded from the analysis, as the beta distribution requires values strictly between 0 and 1.

Colors denote agricultural land-type categories (landtype1 and landtype2). Thick white lines delineate the 14 regional block boundaries, and thin white lines delineate prefectural boundaries (47 prefectures). Map created by the authors in QGIS using public geospatial data from MAFF [39] and Ministry of Land, Infrastructure, Transport and Tourism [40] (see Methods for sources and licensing).

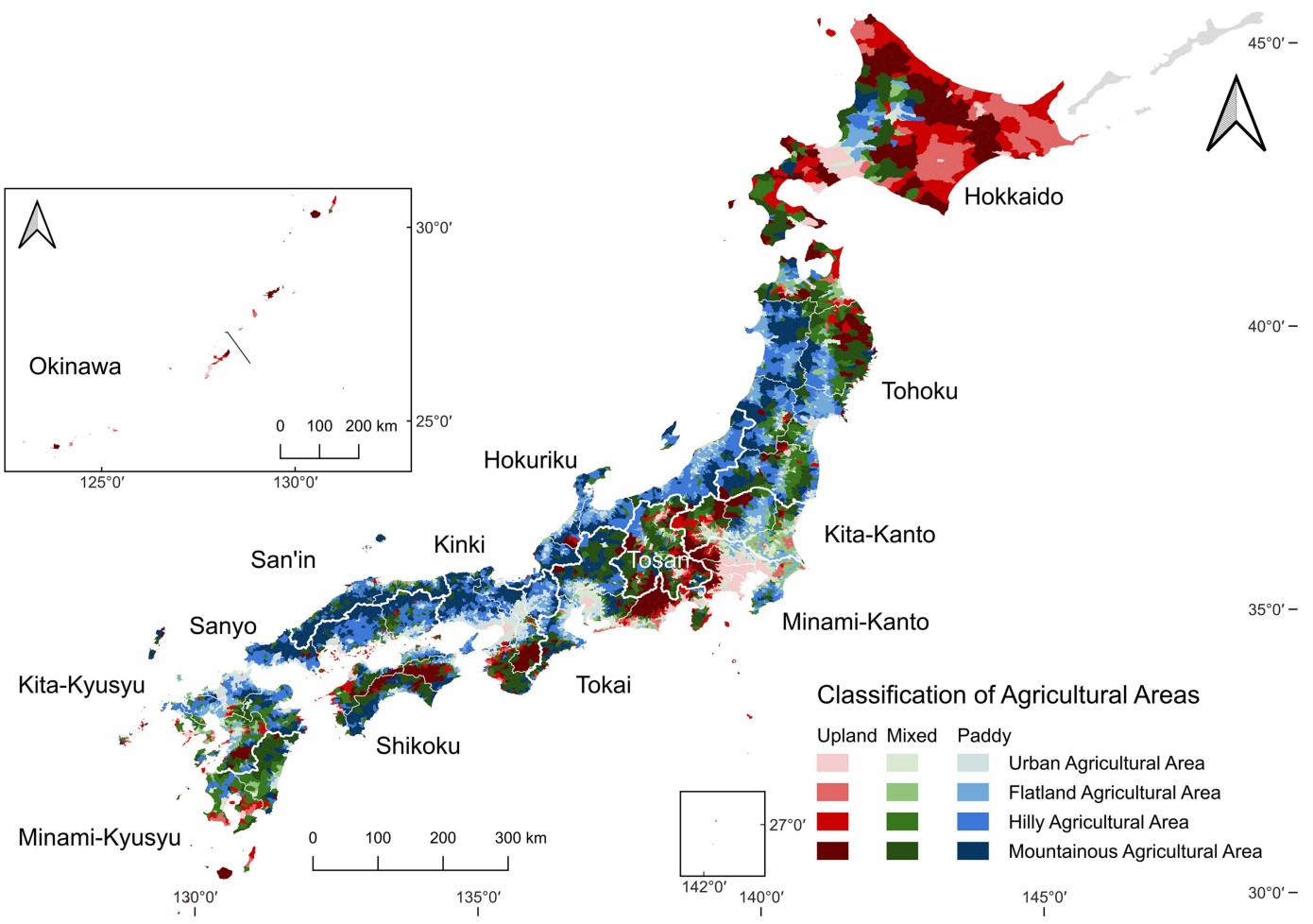

**Fig 1. Japan's agricultural land-type and regional block divisions.**

To identify spatial hotspots and coldspots of agricultural decline and aging, we computed a distance-weighted neighborhood mean for each indicator for every sub-municipality using a distance-weighted neighborhood within 50 km of each polygon centroid. Distances were measured between centroids, and we applied a linear distance-decay weighting truncated at 50 km. This neighborhood mean was then standardized across all sub-municipalities to obtain a local neighborhood Z-score. We defined hotspots and coldspots using percentile cutoffs of the empirical national distribution for each indicator and year. Hotspots were areas with local neighborhood Z-scores at or above the 90th percentile (top 10%), and coldspots were those at or below the 10th percentile (bottom 10%). Accordingly, a hotspot for the decline rate marks areas where the decrease in core agricultural workers is particularly severe, whereas a coldspot indicates areas with only small decreases or even increases. For the proportion of the elderly, hotspots denote areas with especially high proportions of residents aged 75 years or above, whereas coldspots indicate areas where this proportion is relatively low. All statistical analyses were performed in R v4.5.0 [41] using the packages glmmTMB [42], emmeans [43], and DHARMs [44]. Maps were produced in QGIS (v3.40.5) using administrative boundary data from MAFF [39], in accordance with the MAFF Terms of Use [45] and the Public Data License (PDL) 1.0 [46], and using data from the National Land Numerical Information download service of the Ministry of Land, Infrastructure, Transport and Tourism (MLIT) [40], in accordance with the MLIT download service terms of use [47], which state compatibility with CC BY 4.0.

## Results

### Spatial distribution and regional trends in workforce decline (2005–2020)

Fig 2 shows the absolute change in the number of core agricultural workers across the 14 regional blocks between 2005 and 2020. All regions experienced a decline in the number of workers during this period, underscoring the nationwide nature of agricultural depopulation. Among these, the Tosan, Kinki, and Sanyo regions recorded the accelerating decline over time

The horizontal axis shows the 14 regional blocks and Agricultural Census years (2005, 2010, 2015, 2020); the vertical axis shows the number of core agricultural workers.

When examining the percentage decline at the sub-municipality level (Fig 3), it became evident that the extent of workforce reduction varied substantially spatially. While most regions exhibited some degree of decline, the degree of spatial heterogeneity was pronounced. For example, in parts of Kinki, Sanyo, and Tohoku, the number of workers remained stable or even increased in localized areas, whereas regions such as Hokkaido, Kita-Kanto, Tokai, and Minami-Kyushu showed almost uniformly negative trends, with nearly all sub-municipalities experiencing workforce reductions. Particularly in southeastern Tohoku, some areas experienced a near-total loss of farmers. This finding is plausibly attributable to the aftermath of the 2011 nuclear accident, which resulted in long-term evacuation zones and the discontinuation of farming activities.

The map shows the percentage decline in the number of core agricultural workers in 2020 relative to the 2005 baseline. Colors indicate decline-rate classes (see legend). Areas where the 2020 count exceeded the 2005 count are categorized as an Increase. Sub-municipalities with missing data are shown in gray. The number of sub-municipalities with valid data was 10735. Map created by the authors in QGIS using public geospatial data from MAFF [39] and MLIT [40] (see Methods for sources and licensing).

### Changes in the aging structure of core agricultural workers (2005–2020)

Figs 4 and 5 depict the spatial distribution of the proportion of core agricultural workers aged 75 years and above in 2005 and 2020, respectively. In 2005, aging levels showed substantial regional variation. The Tokai, Kinki, San'in, and Sanyo regions showed relatively high aging rates, with the elderly proportions exceeding 30% in many sub-municipalities. In some extreme cases, the fraction of elderly workers surpassed 50%, indicating an already critical demographic state in parts of western Japan. In contrast, regions such as Hokkaido, Tohoku, Kita-Kyushu, Minami-Kyushu, and Okinawa

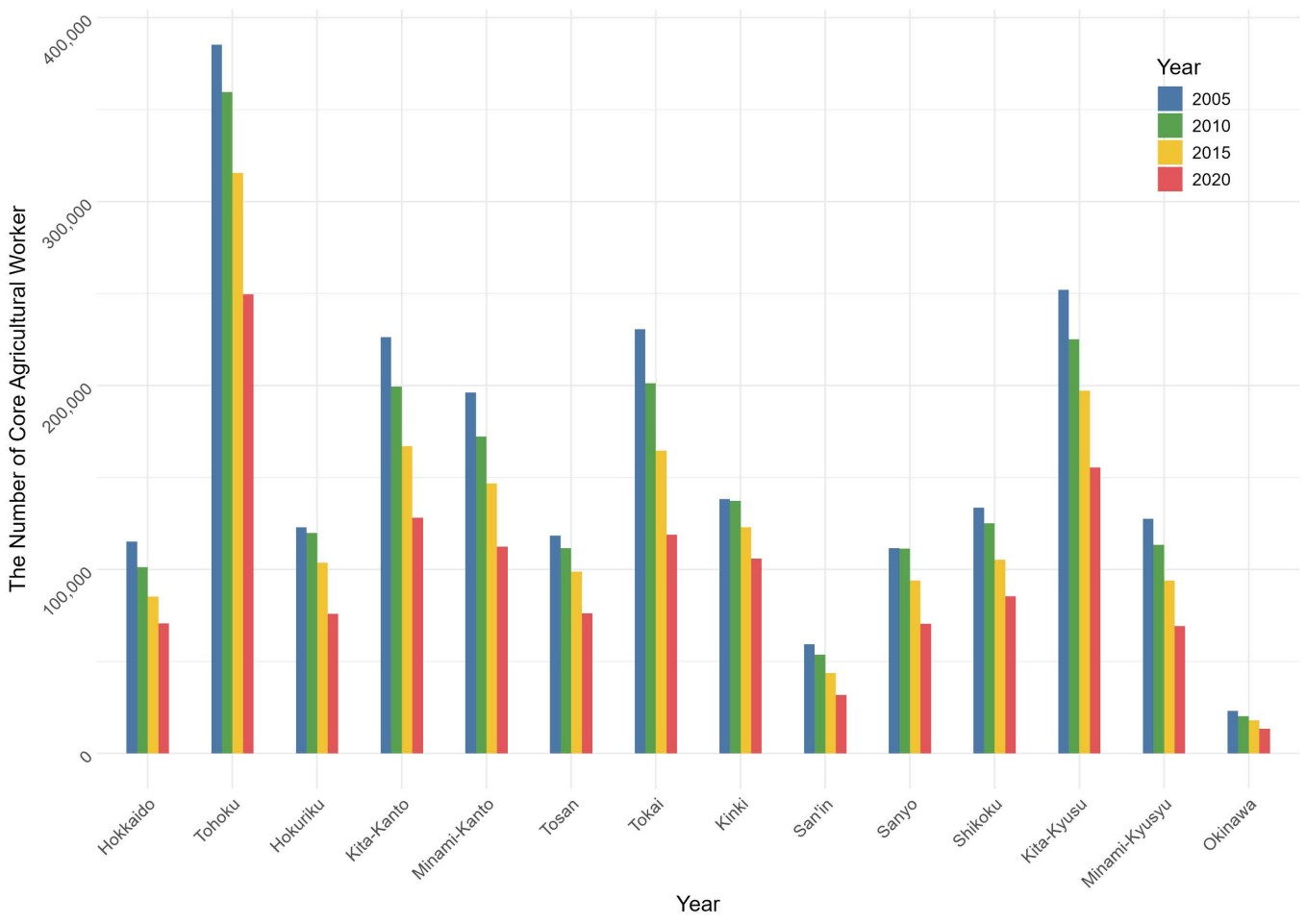

**Fig 2. Trends in the number of core agricultural workers.**

exhibited relatively low aging rates. Particularly in eastern-central Hokkaido, which encompasses large-scale farming areas, several municipalities recorded aging rates below 10%, highlighting a concentration of younger agricultural workers at that time.

The map shows the proportion of core agricultural workers aged 75 years and over in 2005. Colors indicate percentage classes (see legend). Sub-municipalities with missing data are shown in gray. The number of sub-municipalities with valid data was 11059. Map created by the authors in QGIS using public geospatial data from MAFF [39] and MLIT [40] (see Methods for sources and licensing).

The map shows the proportion of core agricultural workers aged 75 years and above in 2020. Colors indicate percentage classes (see legend). Sub-municipalities with missing data are shown in gray. The number of sub-municipalities with valid data was 10858. Map created by the authors in QGIS using public geospatial data from MAFF [39] and MLIT [40] (see Methods for sources and licensing).

By 2020 a marked nationwide shift was evident (Fig 5). In numerous sub-municipalities across the country, workers aged 75 years and above accounted for more than 50%, with particularly high values observed in Tosan, Tokai, Kinki, San'in, Sanyo, and Shikoku. Even in Hokkaido, which had previously maintained lower aging levels, a clear upward trend was observed by 2020.

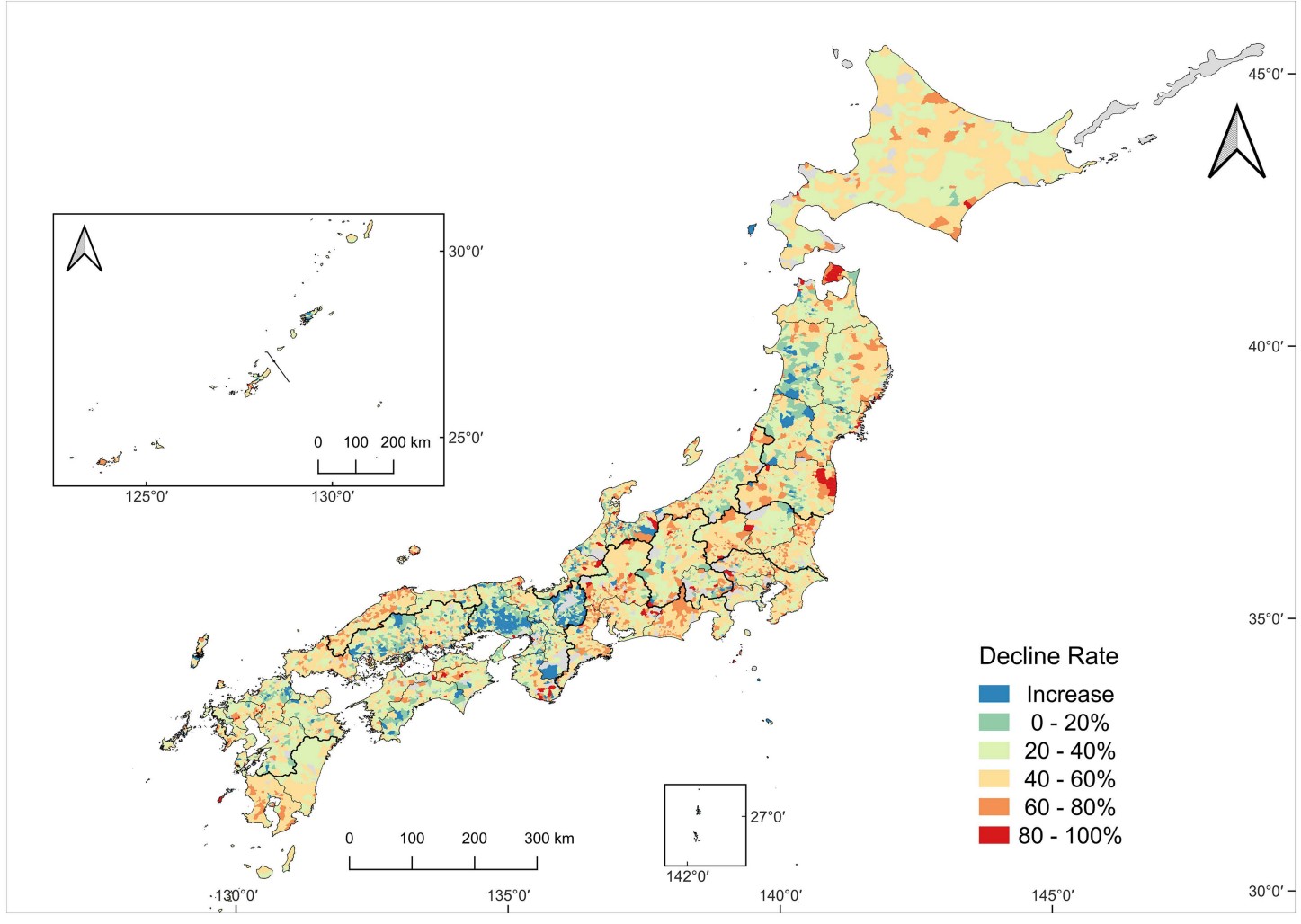

**Fig 3. Decline rate of core agricultural workers (2005–2020).**

## Effects of land-use type

Fig 6 displays model-predicted means (estimated marginal means [EMMs], response scale) from beta-logit GLMMs with a random intercept for region, for all combinations of topographic category (Urban, Flatland, Hilly, Mountainous Agricultural areas; landtype1) and land-use type (Paddy-dominated, Paddy-Upland-Mixed, and Upland-dominated Agricultural areas; landtype2). Table 1 summarizes model fit and omnibus test results, indicating significant effects of landtype1 and of landtype1 × landtype2 nesting for both outcomes. Numerical EMMs and pairwise contrasts are provided in S1 and S2 Tables (S1 File).

Estimated marginal means (response scale) from beta–logit GLMMs are shown for the (a) decline rate (2005–2020) and (b) proportion of the elderly (≥75 in 2020) across all combinations of landtype1 (vertical axis: Urban, Flatland, Hilly, Mountainous Agricultural Areas) and landtype2 (horizontal axis: Paddy-dominated, Paddy-Upland-Mixed, Upland-dominated Agricultural Areas). The 14 regional blocks are modeled as a random intercept. See Table 1 for omnibus tests and S1–S2 Tables (S1 File) for numerical EMMs and contrasts.

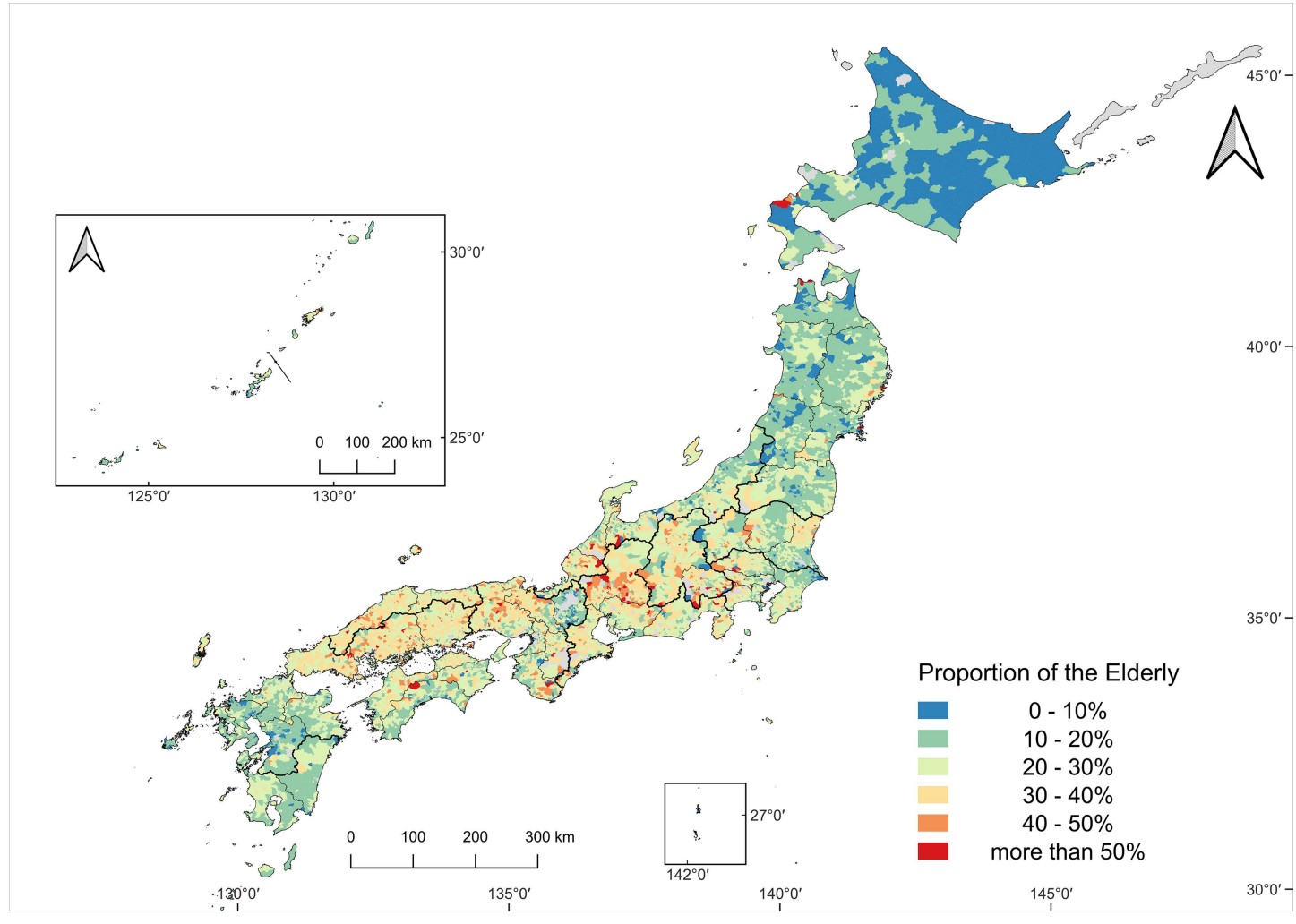

**Fig 4. The proportion of the elderly among core agricultural workers in 2005.**

Outcomes are the decline rate and proportion of the elderly among core agricultural workers. Both models use a beta distribution with a logit link and include a random intercept for region blocks (14 groups). "Dispersion ($\varphi$)" is the estimated beta dispersion parameter. "Random intercept SD (region block)" is the standard deviation of the regional random effect. Omnibus tests (likelihood-ratio tests) compare nested models: landtype1 vs. the intercept-only model ($\chi^2$ df = 3), and landtype1:landtype2 vs. the landtype1 model ($\chi^2$ df = 8). Land types were defined according to the MAFF [34]. Landtype1 included Urban, Flatland, Hilly, and Mountainous Agricultural Area. Landtype2 included Paddy-dominated, Paddy-Upland-Mixed, and Upland-dominated Agricultural Area. Models encode the nested structure as landtype1/landtype2 ≡ landtype1 + landtype1:landtype2; no main effect of landtype2 is included. Lower AIC (Akaike's Information Criterion) indicates better fit. Reference categories and full fixed-effect coefficients are provided in the Supporting Information S2 Appendix (S1 File).

As for the decline rate, the lowest predicted means occurred in Flatland-Paddy and Flatland-Upland combinations, which did not significantly differ from each other, whereas replacing paddy with Upland within Hilly, Mountainous, and Urban areas resulted in a higher decline rate (Tukey-adjusted contrasts, S2 Table in S1 File). The Mountainous-Upland

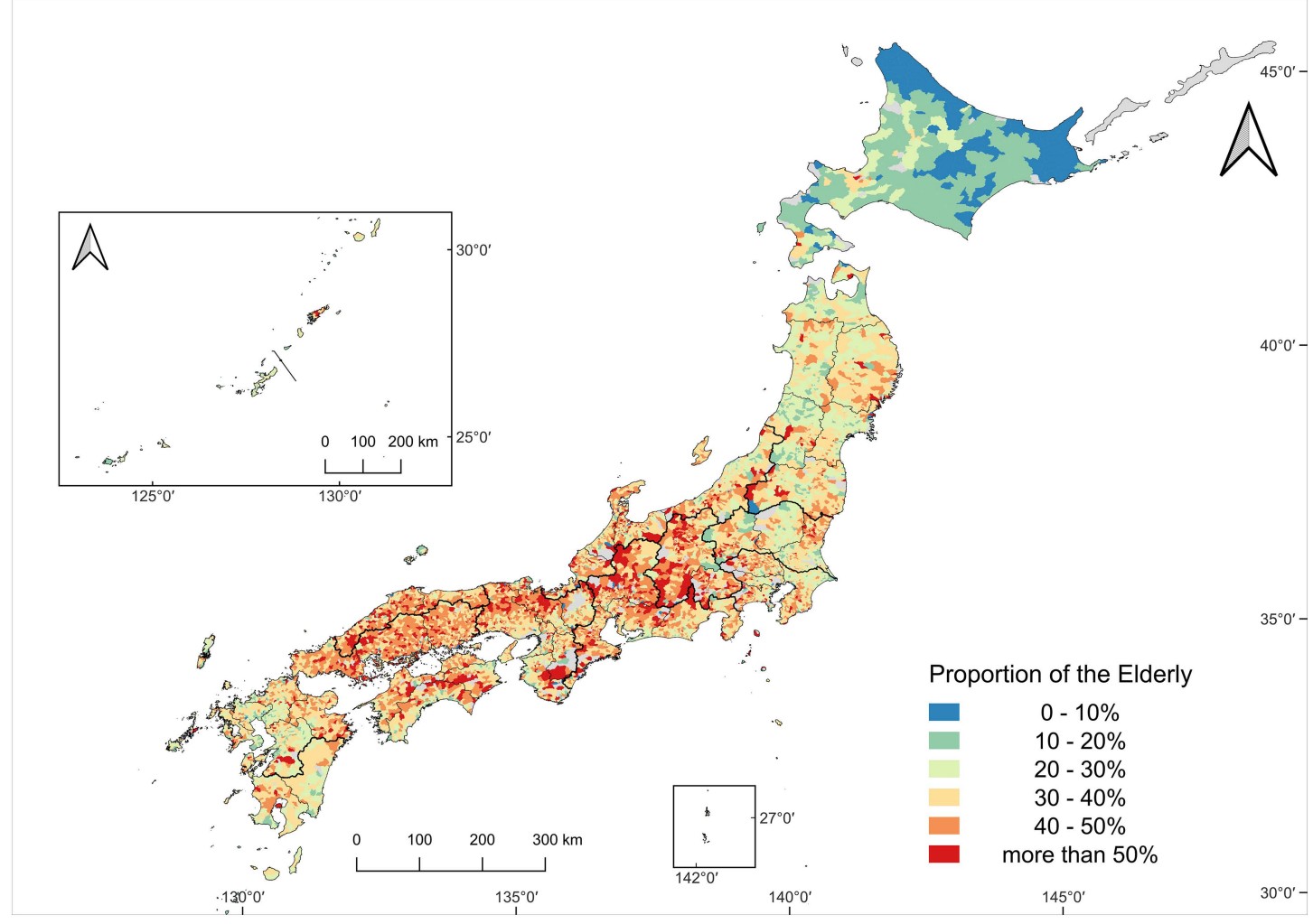

**Fig 5. The proportion of the elderly among core agricultural workers in 2020.**

combination showed the highest decline among all cells, and all Paddy-Upland-Mixed combinations were associated with higher declines than the Flatland-Paddy reference.

As for the proportion of the elderly, the Flatland-Upland combination yielded the lowest predicted mean. Across all topographical categories, Upland had lower elderly proportions than Paddy (within-landtype1 contrasts significant in S2 Table in S1 File), whereas Paddy-Upland-Mixed tended to be similar to Paddy in terms of elderly proportion, with only small and non-significant differences.

Taken together, these findings indicate that both outcomes depend strongly on the interaction between topography and land-use type, as visualized in Fig 6 and supported by the omnibus test results in Table 1 and EMM/contrast summaries in S1 and S2 Tables (S1 File).

## Hotspot and coldspot patterns in the agricultural workforce decline and aging

Hotspot–coldspot mapping revealed distinct spatial patterns for the decline rate (Fig 7) and proportion of the elderly (Fig 8). Because distance-weighted local averaging can be sensitive to sparse neighborhoods and edge effects

## (a) Decline Rate

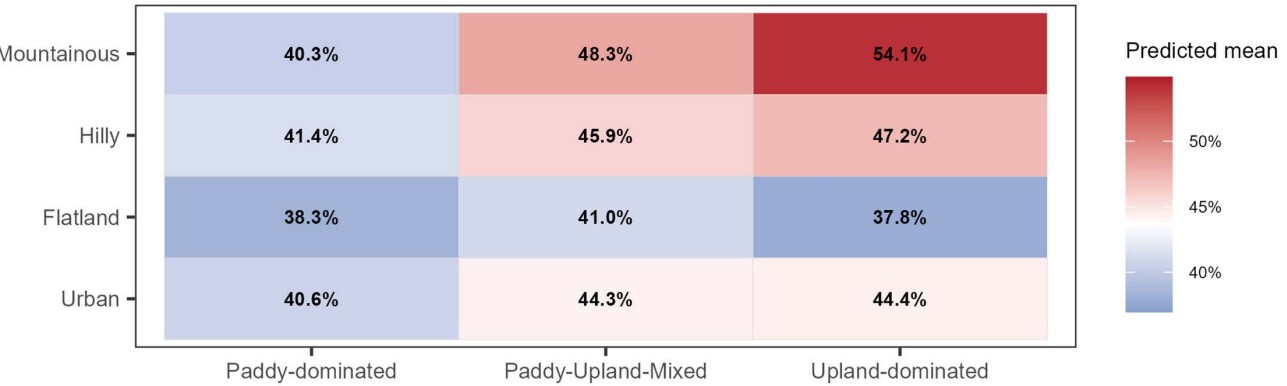

## (b) Proportion of the Elderly

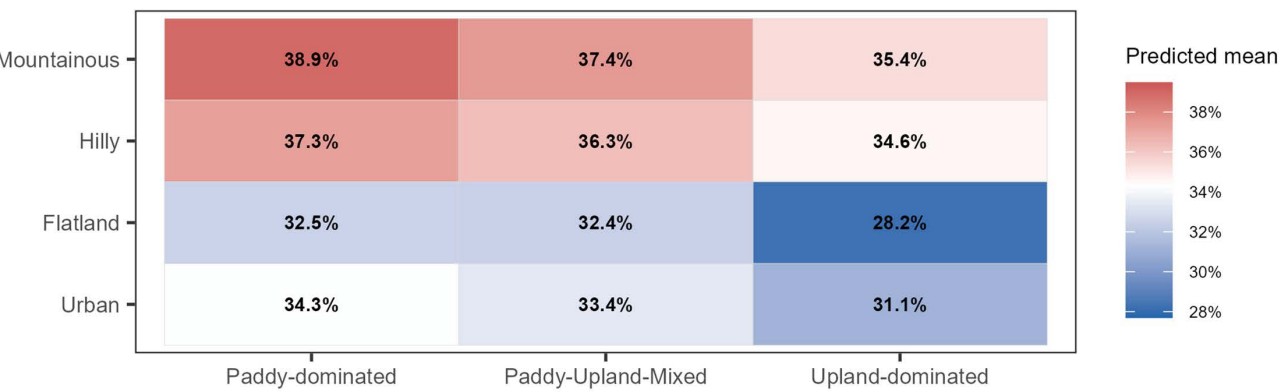

**Fig 6. GLMM-predicted means by land-use category (decline rate and aging rate).**

**Table 1. Summary of beta–logit GLMMs for decrease and elderly rates with a regional random intercept.**

| Model | Family-Link | N | Groups (region block) | AIC | Dispersion ($\varphi$) | Random intercept SD (region block) | Omnibus tests (LRT) | | | |
|---|---|---|---|---|---|---|---|---|---|---|
| | | | | | | | Landtype1 | | Landtype1:Landtype2 | |
| | | | | | | | $\chi^2(3)$ | p | $\chi^2(8)$ | p |
| Decline | Beta-logit | 10172 | 14 | −4422.8 | 4.76 | 0.2083 | 128.67 | < 0.001 | 169.18 | < 0.001 |
| Elderly | Beta-logit | 10854 | 14 | −9487.5 | 7.72 | 0.2965 | 199.149 | < 0.001 | 53.881 | < 0.001 |

on remote islands, we confined our interpretation to Japan's four main islands (Hokkaido, Honshu, Shikoku, and Kyushu).

Hotspot–coldspot map based on a 50-km distance-weighted local average of the decline rate (2005–2020), identified nationwide. Areas in the top decile (upper 10%) are classified as hotspots and those in the bottom decile (lower 10%) as coldspots. Colors indicate hotspot/coldspot classes (see legend). Map created by the authors in QGIS using public geo-spatial data from MAFF [39] and MLIT [40] (see Methods for sources and licensing).

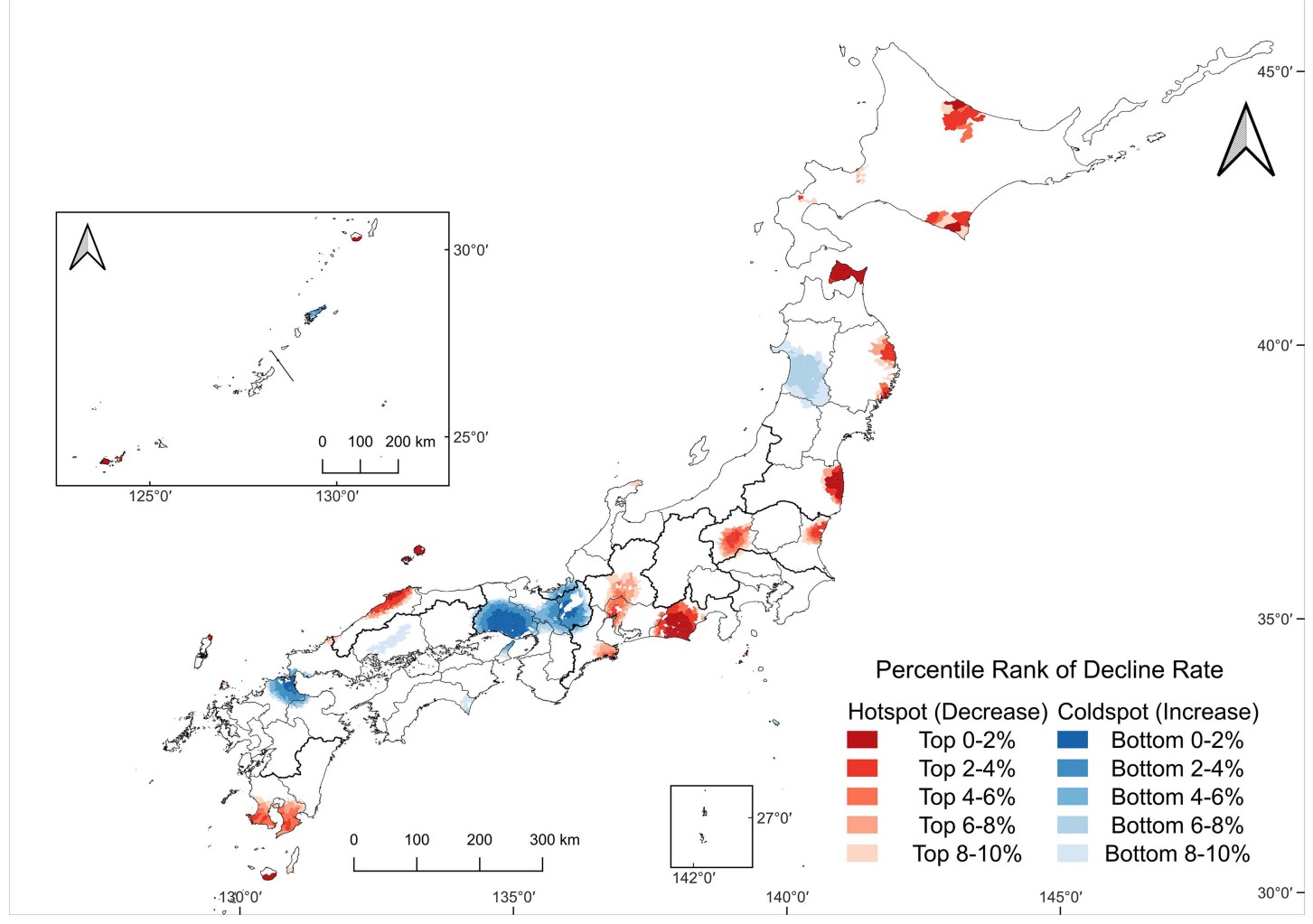

**Fig 7. Hotspots and coldspots of the decline rate.**

Hotspot–coldspot map based on a 50 km distance-weighted local average of the proportion of the elderly (share of core agricultural workers aged ≥75 in 2020), identified nationwide. Areas in the top decile (upper 10%) are classified as hotspots and those in the bottom decile (lower 10%) as coldspots. Colors indicate hotspot/coldspot classes (see legend). Map created by the authors in QGIS using public geospatial data from MAFF [39] and MLIT [40] (see Methods for sources and licensing).

For the decline rate (Fig 7), hotspots were scattered nationwide in Hokkaido, the northernmost and southeastern parts of Tohoku, Kita-Kanto, Tokai, San'in, and Minami-Kyushu. Declining coldspots were found along the western side of Tohoku, Kinki, and Kita-Kyushu regions. For the proportion of the elderly (Fig 8), three principal hotspot areas emerge: (i) a mountainous belt from the Tosan region into Tokai, (ii) the northern part of Kinki, and (iii) a zone spanning from Sanyo to San'in. Elderly coldspots were the most evident in Hokkaido and broadly across Tohoku.

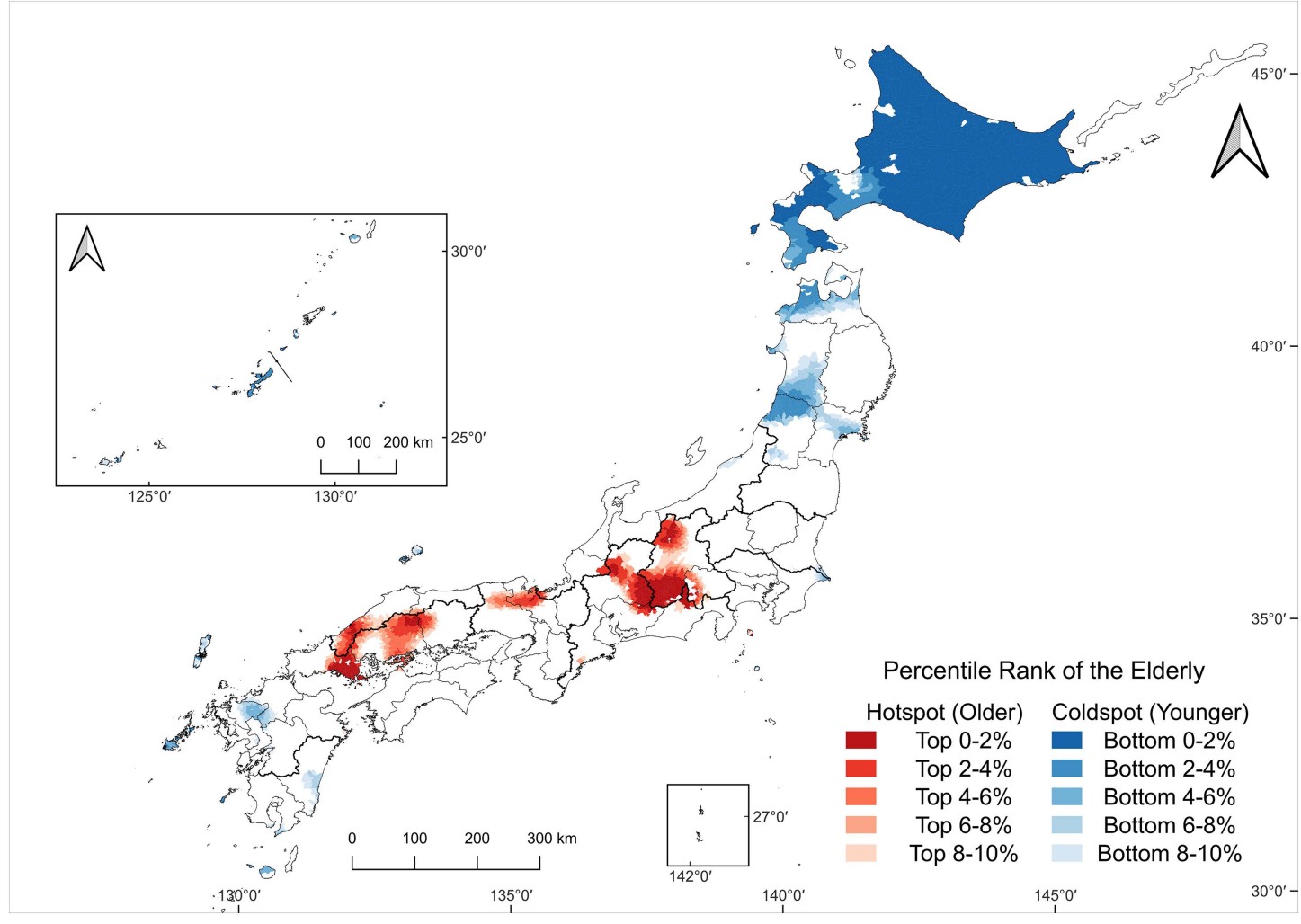

**Fig 8. Hotspots and coldspots of the elderly.**

## Discussion

### Regional disparities and structural drivers of agricultural workforce decline

The study results indicated that the decline in the number of core agricultural workers is a nationwide trend, observed across all 14 regional blocks. While there are some localized areas of workforce growth, particularly in the Kinki and Kita-Kyushu regions, isolated cases are rather than a widespread pattern, the overall trend of workforce reduction is consistent throughout the country.

Regions with especially high decline rates tend to be characterized by mountainous areas and upland-dominated areas (Fig 6). Compared with paddy-based rice cultivation in flatland areas, farming in these regions tends to require greater physical labor because it is less mechanized. This reflects the fact that rice has long been the central crop in Japan, and agricultural technology development has historically focused on rice production. Moreover, in mountainous areas, steep slopes and narrow farm roads hinder the use of heavy machinery that is common in large-scale flatland farming. Therefore, population decline in these regions can be interpreted as a direct consequence of aging, as the continuation of labor-intensive farming may become increasingly unfeasible for elderly farmers [48,49].

Interestingly, the few regions that exhibited an increase in agricultural workers were concentrated in the peri-urban zones surrounding Japan's major metropolitan areas, specifically, the Kinki metropolitan fringe, part of the nation's second-largest urban agglomeration, and the Kitakyushu-Fukuoka metropolitan fringe, which surrounds the fourth-largest urban area (Fig 3). Proximity to consumer markets likely plays a role in attracting or retaining agricultural workers in these areas [50]. In contrast, similar effects were not observed in the Kanto metropolitan fringe (Minami-Kanto region, the nation's largest) or Chukyo metropolitan fringe (Tokai region, the third largest), suggesting that agricultural viability in these regions is influenced by a more complex interplay of factors such as income disparities with other industries, land prices, and urban labor market competition [51].

Within the four main islands, several hotspot regions for workforce decline likely reflect distinct structural factors (Fig 7). First, in southeastern Tohoku, the pronounced decline is consistent with the long-term evacuation, land-use disruption, and access restrictions following the 2011 Fukushima nuclear disaster [52]. Second, some hotspots coincide with areas where non-agricultural industries, especially the automotive sector, are concentrated, notably in the Tokai and Kita-Kanto regions. While urban out-migration is nationwide, in these regions, local manufacturing may be interpreted as having absorbed workers who might otherwise have remained in agriculture. Third, regions with persistent geographic constraints, including the northeastern Tohoku, Hokkaido, San'in, and Minami-Kyushu regions, are characterized by remoteness, rugged terrain, poor infrastructure, and limited market access, conditions that could accelerate agricultural depopulation in those areas [53].

## Accelerating aging and the erosion of agricultural sustainability

The aging of core agricultural workers has intensified between 2005 and 2020, with many regions now exhibiting proportions of elderly workers (aged ≥ 75 years) exceeding 50%. Compared with the fact that workers aged ≥75 accounted for only about 3.5% of all employed persons in 2020 [37], the proportion of elderly workers in agriculture has already reached an extremely high level. Although aging was prominent in western Japan as early as 2005, the 15-year trend indicates that this demographic shift has now become a nationwide trend. However, it is important to note that the present analysis was based solely on data on core agricultural workers within individual farm management units. It excluded workers employed in agricultural corporations and cooperatives, where younger individuals, particularly those without inherited farmland, may be entering the sector. Hence, the aging rates reported here may overstate the actual aging level of the entire agricultural workforce.

Given Japan's demographic structure, reduced entry of younger workers represents a broad background tendency toward aging in agriculture, although the extent of this tendency varies across regions. Under this demographic backdrop, a key determinant of spatial differences is whether (and how) farming can be sustained by older workers and whether local conditions can facilitate the entry or retention of younger workers. The analysis also revealed that aging tends to be more pronounced in paddy-dominated areas than in upland-dominated areas, and in mountainous areas versus flatland areas. In many paddy areas, older farmers may remain in production because rice cultivation is supported by specialized machinery and infrastructures, enabling labor-saving and efficient operations even when generational replacement is limited. In contrast, in mountainous areas, aging and population decline appear to co-occur. Harsh economic conditions stemming from labor-intensive farming, far from major consumer markets, and a lack of successors are likely to constrain both younger entry and continued farming by older workers, accelerating both demographic and structural declines [54].

Even in paddy-dominated areas, where elderly farmers remain active, long-term sustainability is questionable. Given that many of these farmers are already over 75 years of age, expecting them to continue farming for another decade is unrealistic. Therefore, areas with a high proportion of the elderly today are likely to face a steep decline in the workforce within the next decade. This has direct implications for food security because paddy-dominated regions underpin staple rice production, and workforce contraction in these areas could reduce domestic supply resilience.

Conversely, regions with relatively low aging rates, such as Hokkaido and the western coastal areas of the Tohoku region, are characterized by expansive and capital-intensive agricultural zones. In these regions, large-scale operations and a high degree of mechanization may have facilitated the entry of younger farmers. Thus, capital investment and operational efficiency could serve as key drivers in mitigating aging trends and sustaining agricultural viability.

## Future prospects for Japanese rural communities and policy implications

This study demonstrated that the dual challenges of agricultural workforce decline and aging have reached critical levels in most Japanese regions. These trends pose not only a threat to the continuity of agricultural production but also to the broader sustainability of rural communities and settlement systems that rely heavily on agricultural activity.

The first dimension of this crisis concerns the viability of agricultural production. As discussed in the previous section, one possible policy direction is to promote further capital investment, large-scale mechanization, and other structural reforms, and to increase income prospects to attract younger generations to farming, particularly in regions with favorable conditions for such transitions. This approach may offer a realistic pathway for sustaining agricultural output in selected areas [55]. However, this may also intensify spatial concentration of domestic supply and increase vulnerability to region-specific shocks, such as extreme weather or earthquakes [56].

The second dimension, i.e., the erosion of rural community functions, is potentially even more severe. In Japan, settlements or municipalities where more than 50% of the population is aged 65 years or above are often referred to as Genkai Shuraku (marginal villages). The demographic profile of Japan's core agricultural workers, with many areas exceeding this threshold for working farmers, suggests that several regions have already moved beyond the stage of a typical depopulated village. In such areas, the concern is not merely about agricultural productivity but about the survival of the settlement itself [19,20].

Importantly, in many mountainous and depopulated areas of Japan, the maintenance of settlement functions, such as road repair at the ends of infrastructure networks, irrigation channel upkeep, and the management of rice paddies and uplands that buffer residential spaces from forests and mountains, has long depended on the voluntary and community-based efforts made by agricultural workers [57,58]. These tasks are essential to sustaining the integrity of rural settlement environments. Hence, the viability of a community is not only determined by the demographic condition of the general population or local governments, but also by the "limit" already reached by agricultural laborers themselves. In many areas, the weakening of the farming population may disrupt community functionality far earlier than anticipated by municipalities or central government authorities. In other words, the actual tipping point of rural collapse may arrive well before it is statistically recognized.

## Methodological contributions and potential for broader application

This study demonstrated the value of reconstructing consistent spatial units based on sub-municipalities and integrating spatially explicit demographic data with statistical modeling and hotspot analysis to identify fine-scale variations in agricultural sustainability. In Japan, where large-scale municipal mergers have significantly complicated the tracking of long-term trends using official statistics, this approach restores temporal continuity in national-scale datasets and provides a replicable framework for longitudinal rural analysis.

Importantly, this study represents one of the first systematic attempts to visualize shifts in rural demographic dynamics across the entire Japanese archipelago from 2005 to 2020 at the sub-municipality level, producing detailed maps that reveal fine-grained spatial patterns of agricultural population decline and aging.

While this study focused on the number and aging of core agricultural workers, the same methodology can be extended to other census-based agricultural indicators, such as sales value, cultivated land area, and production type. Such

extensions would enable a more comprehensive understanding of both the current status of local agricultural systems and the structural factors shaping them over time. This, in turn, could inform future assessments of regional sustainability and help to identify policy-relevant drivers of agricultural change.

Furthermore, the administrative codes employed in this study align with those employed in other major national surveys, including the Population Census and Economic Census. Hence, the methodological framework can be applied to non-agricultural datasets. Such cross-sectoral analysis could further deepen our understanding of how agricultural, demographic, and economic trends interact at the local level, and how these interactions influence rural sustainability in Japan as well as other countries facing similar challenges.

## Supporting information

**S1 File. Supporting appendices and tables.** This file contains: S1 Appendix, land-type definitions and classification procedures; S2 Appendix, additional information on the methods used; S1 Table, estimated marginal means (response scale) by land-type combination for population decrease and elderly proportion; and S2 Table, Tukey-adjusted pairwise contrasts among the 12 land-type combinations (link scale).
(DOCX)

## Acknowledgments

We are grateful to Dr. Sawada of the Central Region Agricultural Research Center, NARO, and Dr. Teratani of the Department of Agricultural Management Strategy, Headquarters for Management Strategy, NARO, for their valuable advice on the latest insights from the Japanese Census of Agriculture and Forestry. We also thank Dr. Kubotera of the Institute of Agro-Environmental Sciences, NARO, and Dr. Nagai of the Institute of Agro-Environmental Sciences, NARO, for their helpful guidance on the methodology and interpretation of the data.

## Author contributions

**Conceptualization:** Kazuho Funakawa, Toshihiro Sakamoto, Kohei Imamura, Mizuki Morishita, Shoji Taniguchi, Nobusuke Iwasaki, Gen Sakurai.

**Data curation:** Kazuho Funakawa, Toshihiro Sakamoto, Kohei Imamura, Mizuki Morishita, Shoji Taniguchi, Nobusuke Iwasaki, Gen Sakurai.

**Formal analysis:** Kazuho Funakawa.

**Funding acquisition:** Toshihiro Sakamoto, Kohei Imamura, Mizuki Morishita, Shoji Taniguchi, Nobusuke Iwasaki, Gen Sakurai.

**Investigation:** Kazuho Funakawa.

**Methodology:** Kazuho Funakawa, Toshihiro Sakamoto.

**Project administration:** Toshihiro Sakamoto, Gen Sakurai.

**Supervision:** Toshihiro Sakamoto, Gen Sakurai.

**Validation:** Kazuho Funakawa.

**Visualization:** Kazuho Funakawa.

**Writing – original draft:** Kazuho Funakawa, Toshihiro Sakamoto, Kohei Imamura, Mizuki Morishita, Shoji Taniguchi, Nobusuke Iwasaki, Gen Sakurai.

**Writing – review & editing:** Kazuho Funakawa.

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
