## [Decision Letter · Decision Letter 0]

6 Feb 2026

Dear Dr. Funakawa,

Thank you for submitting your manuscript to PLOS ONE. After careful consideration, we feel that it has merit but does not fully meet PLOS ONE’s publication criteria as it currently stands. Therefore, we invite you to submit a revised version of the manuscript that addresses the points raised during the review process.

We look forward to receiving your revised manuscript.

Kind regards,

Yasuko Kawahata

Academic Editor

PLOS One

**Journal Requirements:**

“This work was supported by the Environmental Restoration and Conservation Agency (grant number JPMEERF25S12421) awarded to G.S., N.I., T.S., M.M., K.I., and S.T. (https://www.erca.go.jp/erca/english/index.html

). Additional support was provided by the Japan Society for the Promotion of Science (grant number 25K21880) awarded to N.I. (https://www.jsps.go.jp/english/).”

3. We note that Figures  1, 3, 4 5, 7 and 8  in your submission contain map images which may be copyrighted. All PLOS content is published under the Creative Commons Attribution License (CC BY 4.0), which means that the manuscript, images, and Supporting Information files will be freely available online, and any third party is permitted to access, download, copy, distribute, and use these materials in any way, even commercially, with proper attribution. For these reasons, we cannot publish previously copyrighted maps or satellite images created using proprietary data, such as Google software (Google Maps, Street View, and Earth). For more information, see our copyright guidelines: http://journals.plos.org/plosone/s/licenses-and-copyright.

A.  You may seek permission from the original copyright holder of Figures  1, 3, 4 5, 7 and 8   to publish the content specifically under the CC BY 4.0 license.

B. If you are unable to obtain permission from the original copyright holder to publish these figures under the CC BY 4.0 license or if the copyright holder’s requirements are incompatible with the CC BY 4.0 license, please either i) remove the figure or ii) supply a replacement figure that complies with the CC BY 4.0 license. Please check copyright information on all replacement figures and update the figure caption with source information. If applicable, please specify in the figure caption text when a figure is similar but not identical to the original image and is therefore for illustrative purposes only.

4. Please include your tables as part of your main manuscript and remove the individual files. Please note that supplementary tables (should remain/ be uploaded) as separate "supporting information" files.

Reviewers' comments:

Reviewer's Responses to Questions

**Comments to the Author**

1. Is the manuscript technically sound, and do the data support the conclusions?

Reviewer #1: Yes

Reviewer #2: Yes

2. Has the statistical analysis been performed appropriately and rigorously?

Reviewer #1: Yes

Reviewer #2: N/A

3. Have the authors made all data underlying the findings in their manuscript fully available?

Reviewer #1: Yes

Reviewer #2: Yes

4. Is the manuscript presented in an intelligible fashion and written in standard English?

Reviewer #1: Yes

Reviewer #2: Yes

Reviewer #1: This manuscript reports interesting and important results visually. I have no major concerns, but the following should be properly amended

Line 57: Why 2005 and 2020 are selected should be described.

Line 108: Need to define ‘hotspot’.

Lines 221-222: Explain how these topographical categories were determined.

Lines 222-223: Explain how these land-use types were determined. “as defined by the MAFF” is not enough.

Line 232: Explain landtype1 and landtype 2.

Line 260: ‘workforce size’ is not clear.

Lines 312-316: These are not result and should be transferred to Discussion.

Line 351: What is ×?

Line 395: “broadly across Tohoku” is confusable, because you mention that the northernmost and southeastern parts of Tohoku are hotspots.

Reviewer #2: Comments to the Author

1. I have really enjoyed reviewing this manuscript, especially the methodology section. The methodology has a very solid foundation. The detailed explanation of stages taken to arrive at sub-municipalities makes it easy for other researchers to replacate the study in both Japan and beyond.

2. The problem statement is well grounded and persuasively and convincingly shows clear research gap that was being addressed by the study.

3. Even though I have not had access to some of the figures, it is clear from the presentation of the statistics that the statistical analysis was both rigorous and robust.

4. The results are easy to follow even to non-scientists which is good for wider readership.

5. The discussion of the results should be improved in terms of relating the results to previous studies but also linking them to relevant theories. In fact, to me this is the minor but important issue that the authors should consider addressing. In other words, while the study is making very important methodological contributions, it has remained silent on its theoretical underpppings and theoretical contributions.

6. Both conclusions and policy implications are speaking and well connected to results of the study. This is commendable given that it is not uncommon to see research conclusions and policy recommendations that are completely disconnected from research findings. However, it will do the paper more justice if a discussion is made on implications of the results on as far as food security in Japan in concerned.

7. I have attached the manuscript with my comments in track changes.

8. Once again, thank you for a well written paper with excellent use of the standard English language.

.

Reviewer #1: No

Reviewer #2: **Yes:**Masautso Joseph ChimomboMasautso Joseph ChimomboMasautso Joseph ChimomboMasautso Joseph Chimombo

---

## [Author Response · Author response to Decision Letter 1]

4 Mar 2026

Please see the attached “Response to Reviewers” document for our point-by-point responses to all editor and reviewer comments. We have also uploaded a marked-up manuscript showing changes and a clean revised manuscript.

---

## [Decision Letter · Decision Letter 1]

19 Mar 2026

Identifying where Japanese agriculture is most at risk: a longitudinal analytical framework based on municipal boundaries as of 1950 for workforce decline and aging (2005-2020)

PONE-D-25-52270R1

Dear Dr. Kazuho Funakawa,

We’re pleased to inform you that your manuscript has been judged scientifically suitable for publication and will be formally accepted for publication once it meets all outstanding technical requirements.

Kind regards,

Yasuko Kawahata

Academic Editor

PLOS One

Additional Editor Comments (optional):

Reviewers' comments:

Reviewer's Responses to Questions

**Comments to the Author**

Reviewer #1: All comments have been addressed

Reviewer #2: All comments have been addressed

2. Is the manuscript technically sound, and do the data support the conclusions?

Reviewer #1: Yes

Reviewer #2: Yes

3. Has the statistical analysis been performed appropriately and rigorously?

Reviewer #1: Yes

Reviewer #2: Yes

4. Have the authors made all data underlying the findings in their manuscript fully available?

Reviewer #1: Yes

Reviewer #2: Yes

5. Is the manuscript presented in an intelligible fashion and written in standard English?

Reviewer #1: Yes

Reviewer #2: Yes

Reviewer #1: The revised manuscript is judged to be properly amended to the previous comments. Thus, I have no more additional comments.

Reviewer #2: (No Response)

.

Reviewer #1: **Yes:**Koki ToyotaKoki ToyotaKoki ToyotaKoki Toyota

Reviewer #2: **Yes:**Masautso Joseph ChiomboMasautso Joseph ChiomboMasautso Joseph ChiomboMasautso Joseph Chiombo

---

## [Editor Report · Acceptance letter]

PONE-D-25-52270R1

PLOS One

Dear Dr. Funakawa,

I'm pleased to inform you that your manuscript has been deemed suitable for publication in PLOS One. Congratulations! Your manuscript is now being handed over to our production team.

Kind regards,

on behalf of

Dr. Yasuko Kawahata

Academic Editor

PLOS One